# Mitochondrial DNA copy number variation across human cancers

Ed Reznik[1]*, Martin L Miller[2], Yasin Şenbabaoğlu[1], Nadeem Riaz[3],
Judy Sarungbam[4], Satish K Tickoo[4], Hikmat A Al-Ahmadie[4], William Lee[1,3],
Venkatraman E Seshan[5], A Ari Hakimi[1,6], Chris Sander[1]

[1]Computational Biology Program, Memorial Sloan Kettering Cancer Center, New York, United States; [2]Cancer Research UK, Cambridge Institute, Cambridge, United Kingdom; [3]Department of Radiation Oncology, Memorial Sloan Kettering Cancer Center, New York, United States; [4]Department of Pathology, Memorial Sloan Kettering Cancer Center, New York, United States; [5]Department of Epidemiology and Biostatistics, Memorial Sloan Kettering Cancer Center, New York, United States; [6]Urology Service, Department of Surgery, Memorial Sloan Kettering Cancer Center, New York, United States

**Abstract** Mutations, deletions, and changes in copy number of mitochondrial DNA (mtDNA), are observed throughout cancers. Here, we survey mtDNA copy number variation across 22 tumor types profiled by The Cancer Genome Atlas project. We observe a tendency for some cancers, especially of the bladder, breast, and kidney, to be depleted of mtDNA, relative to matched normal tissue. Analysis of genetic context reveals an association between incidence of several somatic alterations, including IDH1 mutations in gliomas, and mtDNA content. In some but not all cancer types, mtDNA content is correlated with the expression of respiratory genes, and anti-correlated to the expression of immune response and cell-cycle genes. In tandem with immunohistochemical evidence, we find that some tumors may compensate for mtDNA depletion to sustain levels of respiratory proteins. Our results highlight the extent of mtDNA copy number variation in tumors and point to related therapeutic opportunities.

*For correspondence: reznike@ mskcc.org

Competing interests: The authors declare that no competing interests exist.

## Introduction

Human cells contain many copies of the 16-kilobase mitochondrial genome, which encodes 13 essential components of the mitochondrial electron transport chain and ATP synthase. Alterations of mitochondrial DNA (mtDNA), via inactivating genetic mutations or depletion of the number of copies of mtDNA in a cell, can impair mitochondrial respiration and contribute to pathologies as diverse as encephelopathies and neuropathies (*El-Hattab and Scaglia, 2013*), and the process of aging (*Balaban et al., 2005*; *Finkel and Holbrook, 2000*). In cancer, a number of studies have examined the role of mtDNA mutations in carcinogenesis (*Wallace, 2012*; *Ju et al., 2014*; *Larman et al., 2012*; *He et al., 2010*). However, the contribution of changes in the gross number of mtDNA genomes in a tumor (i.e. the 'mtDNA copy number') to tumor development and progression has not been adequately investigated.

In contrast to the fixed (diploid) copy number of the nuclear genome, many copies of mtDNA exist within each cell, and these levels can fluctuate. Because mitochondria undergo a constant process of fusion and fission, it is difficult to meaningfully determine the number of mtDNA molecules per mitochondrion. Instead, studies have focused on measuring mtDNA copy number per cell, with estimates for humans that vary between a few hundred and over one hundred thousand copies, depending on the tissue under examination (*Wai et al., 2010*). Furthermore, because mtDNA serves

**eLife digest** Within each cell of your body lie hundreds or thousands of mitochondria. These structures are perhaps best known for making energy, but mitochondria also play roles in processes like the immune response and cell signaling. However, in the mutant cells that form cancerous tumors, these roles can be subverted and altered.

Mitochondria contain their own DNA, which is distinct from the DNA stored in the nucleus of the cell, and codes for the proteins that the mitochondria need to produce energy. Reznik et al. used next-generation DNA sequencing data produced by The Cancer Genome Atlas consortium to estimate the number of copies of mitochondrial DNA in tumor cells and the adjacent normal tissue. This revealed that in many types of cancer, tumor cells have fewer copies of mitochondrial DNA than the cells that make up normal tissue. In many cases, the depletion of mitochondrial DNA was accompanied by a reduction of the expression of mitochondrial genes, suggesting that mitochondrial activity may be suppressed in these tumor types.

Reznik et al. also found that the number of copies of mitochondrial DNA in certain tumor types is related to the incidence of key 'driver' mutations that cause cells to become cancerous. This knowledge may help to develop new treatments for these tumors.

as a template for the transcription of essential electron transport chain complexes, the quantity of mtDNA in a cell may serve a surrogate marker for the cell's capacity to conduct oxidative phosphorylation if the copy number of mtDNA is rate-limiting. For instance, a recent study estimated that energy-intensive tissues such as cardiac and skeletal muscle contained between 4000 and 6000 copies of mtDNA per cell, while liver, kidney, and lung tissues averaged between 500 and 2000 copies (*D'Erchia et al., 2015*).

Mitochondrial dysfunction plays several distinct roles in cancer (*Schon et al., 2012*; *Wallace, 2012*; *Larman et al., 2012*). First, the normal functions of mitochondria (e.g. respiration) may be subverted to support the growth of the tumor. A canonical example of this is the observation that many tumors suppress mitochondrial respiration in favor of increased uptake of glucose and secretion of lactate ('the Warburg effect'), a phenomenon which has found clinical utility for imaging of tumors using FDG-PET (*Vander Heiden et al., 2009*). Second, mitochondria are susceptible to mutations in nuclear- and mitochondrially-encoded genes, and a subset of tumors are known to be caused by mutations of the mitochondrial enzymes FH, SDH, and IDH (*King et al., 2006*). Furthermore, mtDNA dysfunction affecting the electron transport chain can lead to generation of excess reactive oxygen species (ROS), contributing to tumor cell metastasis (*Ishikawa et al., 2008*).

To date, no comprehensive analysis of mtDNA copy number changes in tumors has been completed, despite a large literature of isolated reports (*Yu, 2011*). Large-scale studies of mtDNA in cancer have instead focused on the analysis of mutations and heteroplasmy, largely ignoring the contribution of mtDNA copy number variation to the development and progression of tumors. Here, we use whole-genome and whole-exome sequencing data to examine changes in mtDNA copy number across a panel of cancer types profiled by The Cancer Genome Atlas (TCGA) consortium. Using the resulting mtDNA copy number estimates, we ask fundamental questions about mtDNA and cancer. We investigate whether evidence of the Warburg effect can be found in patterns of mtDNA accumulation or depletion. We further examine the connection between gene expression levels and mtDNA copy number, and identify a subset of mitochondrially-localized metabolic pathways exhibiting a high degree of co-expression with mtDNA levels. Finally, we ask whether gross variations of mtDNA copy number are linked to the incidence of somatic alterations (including mutations and copy number alterations) across cancer types. Altogether, our results shed light on the contribution of aberrant mitochondrial function, through changes in mtDNA content, to cancer.

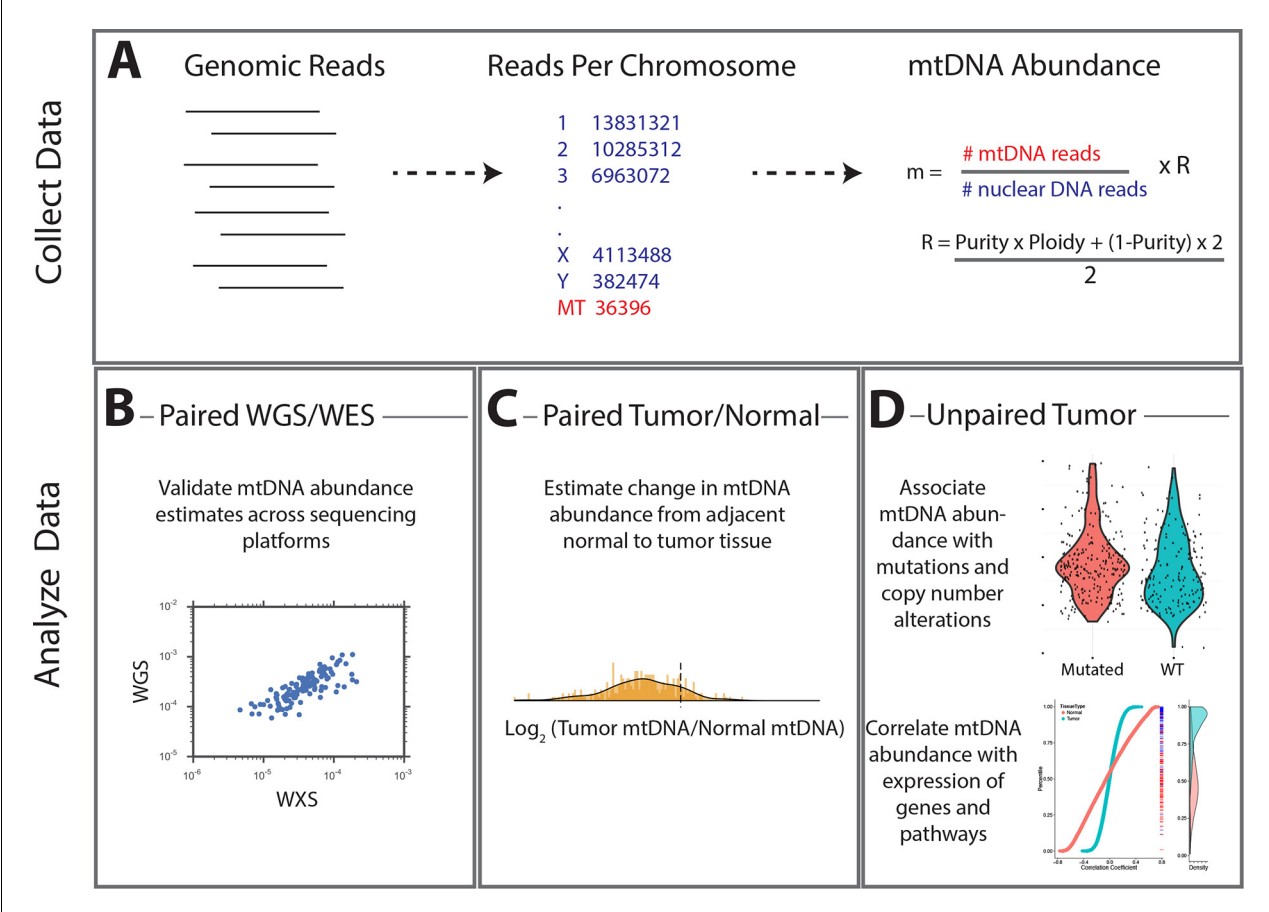

**Figure 1.** Summary of methods. (**A**) Reads were analyzed to determine the number aligning to each chromosome. Relative abundance of mitochondrial DNA was calculated as the ratio of mtDNA reads to nuclear DNA reads, and corrected for tumor purity and ploidy. The results of these calculations were employed in three different types of analysis. (**B**) Comparisons across samples profiled by both whole exome and whole genome sequencing provided validation of mtDNA copy number estimates. (**C**) Pairs of matched tumor/adjacent-normal samples were compared to uncover patterns of mtDNA accumulation and depletion. (**D**) Using all data available (including tumor samples lacking matched normal samples), statistical associations between mtDNA copy number and (1) mutation/copy number alterations, and (2) gene expression, were calculated.

## Results

### Calculation of mtDNA abundance

To estimate the copy number of mtDNA in a tumor sample, we implemented a computationally efficient and fast approach based on comparing the number of sequencing reads aligning to (1) the mitochondrial (MT) genome and (2) the nuclear genome. Comparable approaches have been used to estimate somatic copy number alterations within the nuclear genome in cancer [for a review, see *Zhao et al. (2013)*]. The approach assumes that regions of the genome of equal ploidy should be sequenced to comparable depth. In a normal human cell, the autosomal nuclear genome is at a fixed (diploid) copy number. Thus, by calculating the ratio of reads aligning to the mitochondrial and nuclear genomes, respectively, it is possible to estimate mtDNA ploidy relative to a diploid standard. This approach to assaying mtDNA copy number has been proposed and implemented by others in prior work (*Guo et al., 2013*; *D'Erchia et al., 2015*; *Samuels et al., 2013*).

To estimate mtDNA copy number, we calculated the ratio of (1) the number of sequencing reads mapping to the MT genome ($r_m$) to (2) the number of reads mapping to the nuclear genome ($r_n$) (*Equation 1*). Because tumor cells can exhibit large-scale genomic amplifications and deletions, and may be infiltrated by stromal and immune cells, we applied a ploidy/purity correction ('*R*'), described in detail in the Materials and methods. This calculation yields the relative mtDNA copy number *m*.

Assuming two samples have been processed in identical manners, the sample with a higher value of $m$ contains more copies of mtDNA (*Guo et al., 2013*; *D'Erchia et al., 2015*). In line with previous studies (e.g. [*Ju et al., 2014*]), we observed significant variation in mean mtDNA copy number between sequencing centers, as well as between each batch (i.e., each TCGA plate ID) within a single sequencing center. We applied a batch correction to control for this effect (see Materials and methods).

$$m = \frac{r_m}{r_n} \times R \qquad (1)$$

We applied this method to whole exome sequencing (WXS) and whole genome sequencing (WGS) data from 22 distinct TCGA studies (*Figure 2*, see Materials and methods for further details on data collection). To validate estimates of mtDNA copy number, we compared estimates from samples submitted to both WXS and WGS. Although mitochondrial reads are abundant in both WGS and WXS, the two sequencing methods capture mtDNA at different efficiencies: exome sequencing involves the targeted enrichment of exonic regions prior to sequencing and does not target mtDNA (*Samuels et al., 2013*), while WGS sequences cellular DNA in an unbiased manner. If our approach to estimating mtDNA copy number is accurate, then we expect that the two

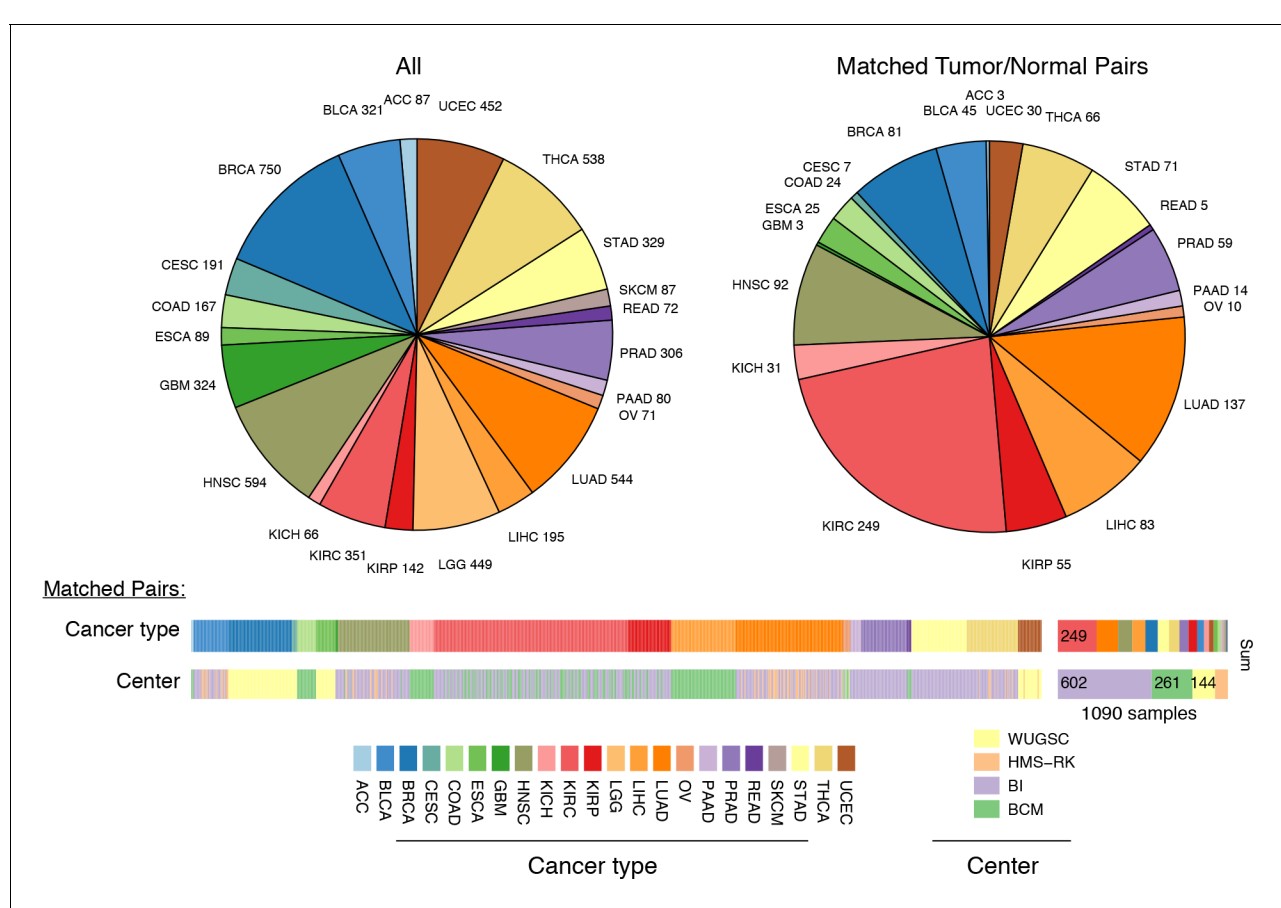

**Figure 2.** Summary of data. Whole-exome and whole-genome sequencing data were obtained from 22 TCGA studies. Abbreviations for each cancer type follow the standard TCGA nomenclature. The data were processed at four different sequencing centers, each of which was analyzed separately. Over 1000 samples were paired instances of tumor/adjacent-normal tissue from the same patient, which were used to quantify changes in mtDNA content across tumors.

The following figure supplement is available for figure 2:

**Figure supplement 1.** Comparison of mtDNA copy number estimates of samples profiled by both whole genome (WGS) and whole exome (WXS) sequencing.

sequencing platforms should offer comparable estimates of mtDNA copy number across a panel of samples, i.e., samples with high mtDNA copy number in WGS should have similarly high mtDNA copy number in WXS. We compared mtDNA copy number estimates in 1110 samples across 8 tumor types profiled by both WXS and WGS, controlling for sequencing center and TCGA plate ID. We confirmed that across all combinations of cancer types and sequencing centers, WXS and WGS offer significantly correlated estimates of mtDNA copy number (*Figure 2—figure supplement 1*).

## Gross changes in mtDNA content are evident in many cancers

Do tumors have different numbers of copies of mtDNA compared to normal tissue? We investigated whether tumor samples showed a significant change in mtDNA content, relative to matched normal tissues. To do so, for each pair of tumor/adjacent-normal samples collected from the same patient, sequenced at a single sequencing center and within the same batch (1090 pairs in total), we calculated the ratio

$$r = \log_2\left(\frac{m_T}{m_N}\right) \tag{2}$$

where $m_T$ and $m_N$ are the mtDNA copy number estimates in tumor and normal tissues, respectively. We then used non-parametric Wilcoxon signed rank tests to assess whether each cancer type was signficantly enriched for tumor samples with higher or lower mtDNA content than matched normal tissue. The analysis was restricted to 15 cancer types for which we had at least 10 matched tumor/normal pairs. To ensure a meaningful comparison, we only used adjacent-normal tissue (and not blood) for the analysis. We elected to focus on analyzing whole-exome sequencing data, for which we had the largest number of samples. A complete list of all calculations is available in *Supplementary file 1*.

Strikingly, seven of the fifteen tumor types analyzed showed a statistically significant (BH-corrected Mann-Whitney p-value <0.05) decrease in mtDNA abundance in tumor samples (*Figure 3A*). These 'mtDNA-depleted' tumor types included bladder (BLCA), breast (BRCA), esophogeal (ESCA), head and neck squamous cell (HNSC), kidney (both clear-cell, KIRC, and papillary, KIRP, but not chromophobe, KICH, subtypes), and liver (LIHC) cancers. Despite a tendency towards mtDNA depletion, all tumor types contained at least one sample with higher mtDNA content than adjacent normal tissue. Nevertheless, the depletion effect was exceptionally strong in several tumor types: except for a handful of WGS samples, nearly all bladder tumors were depleted of mtDNA. Similarly, 87% of clear-cell kidney tumor samples contained less mtDNA than their normal tissue counterparts. In contrast, a single tumor type, lung adenocarcinoma (LUAD), showed statistically significant mtDNA accumulation. In cases where sufficient numbers of both WGS and WXS data were available (bladder, breast, head and neck, clear-cell kidney, thyroid, endometrial, and lung adenocarcinomas), we observed a consistent effect across samples processed by both platforms (*Figure 3—figure supplement 1*).

Our estimates of tumor mtDNA content are based on sequencing DNA of bulk tumor tissue, which includes stromal and immune cell infiltration. While the effect of purity on mtDNA copy number is partially accounted for by the correction factor *R*, it is still possible that tumor impurity may bias the calculation of copy number (for example, if immune cells have lower mtDNA copy number than adjacent-normal tissue). Although our computational method is unable to deconvolve sequencing traces arising from tumor cells vs. infiltrate, we nevertheless investigated the statistical association between tumor mtDNA content and stromal/immune infiltration. We obtained estimates of stromal and immune cell infiltration based on gene expression data for eight tumor types calculated using the ESTIMATE algorithm (*Yoshihara et al., 2013*). We correlated these values with estimates of mtDNA copy number, with full results reported in *Figure 3—figure supplements 2–5* and *Figure 3—source data 1*, with p-values reported as uncorrected for multiple hypothesis testing. We observed a recurrent but weak negative correlation between tumor mtDNA content and immune infiltration score. Further detailed study is required to trace the contribution of infiltrating cells to mtDNA content.

To further relate changes in mtDNA content to clinical progression of disease, we used Cox regression to determine if tumor mtDNA copy number was a significant predictor of patient survival (*Figure 4*). In total, five cancer types showed statistically significant association between patient survival and mtDNA content. In three cancer types (adrenocortical carcinoma, p-value 0.026;

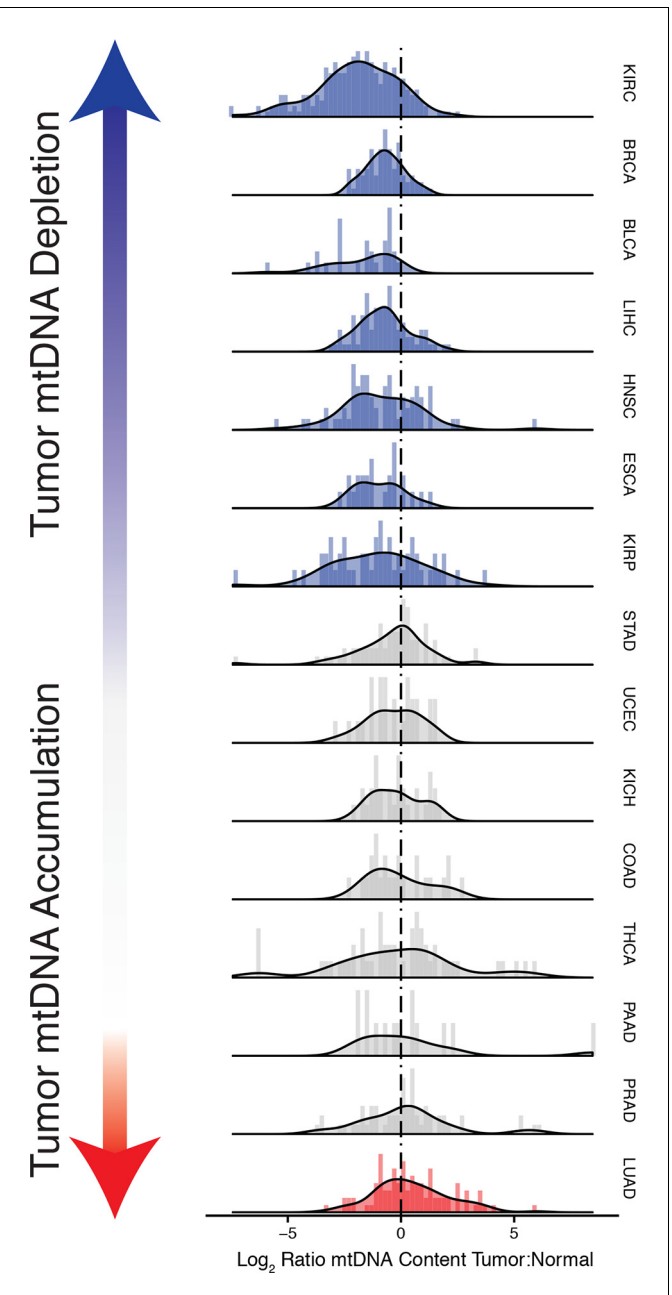

**Figure 3.** Many tumor types show depletion of mtDNA in tumor samples, relative to adjacent normal tissue. Normalized histograms and density plots illustrate log2 ratio of mtDNA content in tumor tissue, to mtDNA content in normal tissue. Each row is a different tumor type. Statistical significance of trends is assessed using a Wilcoxon sign rank test, and p-values are corrected using the Benjamini-Hochberg procedure. Cancer types displaying significant depletion/accumulation of mtDNA are colored in blue/red. Seven of fiteen tumor types show a significant depletion of mtDNA content (a shift of the distribution to the left of the dashed line), relative to normal tissue. One tumor type, lung adenocarcinomas, shows an increase in mtDNA content, relative to normal tissue.

The following source data and figure supplements are available for figure 3:

**Source data 1.** Correlation of mtDNA copy number with ESTIMATE (*Yoshihara et al., 2013*) stromal and immune scores.

**Figure supplement 1.** mtDNA tumor:normal copy number ratio using whole-genome sequencing (WGS) data.

*Figure 3 continued on next page*

*Figure 3 continued*

**Figure supplement 2.** Correlation between tumor mtDNA copy number and ESTIMATE immune scores.

**Figure supplement 3.** Correlation between tumor mtDNA copy number and ESTIMATE stromal scores.

**Figure supplement 4.** Correlation between tumor/normal mtDNA copy number ratio and ESTIMATE immune scores.

**Figure supplement 5.** Correlation between tumor/normal mtDNA copy number ratio and ESTIMATE stromal scores.

chromophobe renal cell carcinoma, p-value 0.053; and low-grade glioma, p-value 0.009), high tumor mtDNA content was associated with better survival. The opposite trend, of poor survival in patients with high tumor mtDNA, was observed in clear-cell renal cell carcinoma (p-value 0.023) and melanoma (p-value 0.043). The finding regarding KICH is particularly intriguing given the central role mitochondrial dysfunction has been proposed to play in the disease (*Davis et al., 2014*). That mtDNA copy number correlates with better or worse survival, depending on cancer type, suggests that other confounding factors strongly tied to survival, such as the presence of somatic mutations, may influence mtDNA levels. In a later section, we will investigate this hypothesis.

## mtDNA copy number is correlated to the expression of mitochondrial metabolic genes

Proteins encoded in mtDNA localize exclusively to the mitochondrial electron transport chain and ATP synthase, and fluctuations in mtDNA copy number are well-known to influence the level of transcription of these genes. It has also been observed that complete depletion of mtDNA in cell lines by exposure to ethidium bromide affects a number of additional signaling pathways (*Chandel and Schumacker, 1999*). Thus, we were compelled to ask if changes in mtDNA content narrowly influenced changes in the expression of oxidative phosphorylation genes, or if they were more broadly connected to the other functions of mitochondria.

Our approach to this question was to search for gene sets whose transcriptional signatures were highly correlated to mtDNA copy number. To do so, we calculated the non-parametric Spearman correlation between the expression of each gene and mtDNA copy number, and then used the mean-rank gene set test implemented in limma (*Law et al., 2014*) to identify gene sets which were significantly enriched for highly correlated genes. The approach was applied in an unbiased manner to all Reactome gene sets in the Canonical Pathways group from the MSigDB database (*Liberzon et al., 2011*).

In general, each tissue exhibited specific gene sets which were strongly correlated to mtDNA copy number levels. However, when aggregating across all cancer types, mitochondrially-localized metabolic pathways showed the most frequent significant correlation with mtDNA abundance (*Figure 5* and *Supplementary file 2*, Worksheet Fig5Data). This recurrent positive correlation between expression of mitochondrial genes and mtDNA copy number across many tumor types served as a second, independent validation that estimates of mtDNA copy number reflected in vivo mtDNA ploidy. We also calculated the correlation between mtDNA copy number and the expression of TFAM, a critical transcription and replication factor which binds to mtDNA in nucleoids, and found a significant positive correlation (Spearman p-value <0.05) in 34% of studies (*Figure 5—source data 1*).

In line with expectation, we found that the 'TCA Cycle and Respiratory Electron Transport' gene set was the most frequently correlated to mtDNA copy number (1st out of 674 gene sets). Among the remaining top positively correlated gene sets, many were metabolism-related, and included mitochondrial beta oxidation of fatty acids, and branched chain amino acid (BCAA) catabolism. BCAAs (valine, leucine, isoleucine) are essential amino acids whose catabolism depends on the activity of an enzyme complex, branched chain $\alpha$-keto acid dehydrogenase, located in the inner mitochondrial membrane (*Hutson et al., 1988*), and prior work has demonstrated that dietary

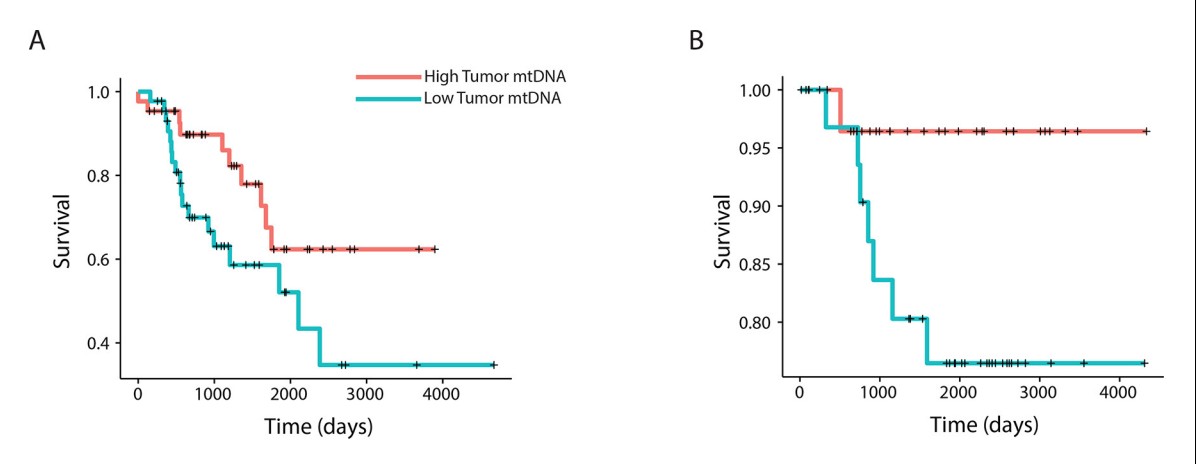

**Figure 4.** mtDNA content is significantly associated with patient survival in (A) adrenocortical (ACC) and (B) kidney chromophobe carcinoma (KICH). For visualization purposes, patients are partitioned into two groups, based on tumor mtDNA copy number relative to the median mtDNA copy number across all tumor samples in the cancer type. Cox regression identified a significant association between high tumor mtDNA and better survival in these two tumor types (ACC, p-value 0.026; KICH, p-value 0.053).

supplementation of BCAA's promotes mitochondrial biogenesis (*Valerio et al., 2011*; *D'Antona et al., 2010*). Furthermore, a recent study has shown that elevated plasma levels of BCAAs are found 2 to 5 years before a cohort of patients developed pancreatic ductal adenocarcinoma (*Mayers et al., 2014*).

A number of gene sets showed recurrent negative correlation to mtDNA copy number (*Figure 5—figure supplement 1* and *Supplementary file 2*, Worksheet Fig5Data). Several of these gene sets, including those related to mRNA processing and the cell cycle, are associated with known non-metabolic functions of mitochondria in the cell. In particular, the replication of mitochondria and mtDNA is intimately linked to the cell cycle (*Chatre and Ricchetti, 2013*), and the nucleotide precursors to mtDNA are in part produced de novo, via a pathway that is only active during the S phase of the cell cycle (*Sigoillot et al., 2003*). Several immune pathways, including those related to interferon signaling, are also frequently negatively correlated with mtDNA content. This is interesting in light of the role that mitochondria play in innate immunity (*West et al., 2011*; *Weinberg et al., 2015*). Of particular interest is a recent report by West and colleagues (*West et al., 2015*), demonstrating that mtDNA stress induced by depletion of TFAM triggered the innate immune response via interferon-stimulated genes and anti-viral signaling. Of the seven tumor types shown to be depleted of mtDNA in *Figure 3*, five (BLCA, BRCA, ESCA, HNSC, KIRC ) exhibit a negative correlation between expression of immune system genes and tumor tissue (but not necessarily normal tissue) mtDNA content.

A subset of tumor types did not show strong positive correlation between mtDNA copy number and expression of mitochondrial metabolic genes. In some cases, this was the result of an apparently dominant correlation with another pathway. Interestingly, in prostate adjacent normal tissue, the expression of mitochondrial respiratory genes was anti-correlated to mtDNA content (see *Supplementary file 2*). We speculate that this effect may be associated with the unique mitochondrial metabolism of prostate epithelia, which secrete large amounts of citrate generated in the mitochondria, rather than oxidizing it further and using the resulting NADH in the respiratory electron transport chain (*Costello et al., 1997*; *2004*).

## Association with mutations and copy number alterations

The landscape of genetic events driving tumors is diverse, and the presence and activity of these genetic lesions is now being used in design of clinical trials and development of new treatments (*Rubio-Perez et al., 2015*). We sought to understand whether mtDNA abundance was associated with the incidence of particular mutations/copy number alterations (CNAs) in patient samples. To do so, we evaluated whether patients with a particular genetic lesion showed statistically significant increases or decreases in tumor mtDNA abundance, compared to wild-type samples. We restricted

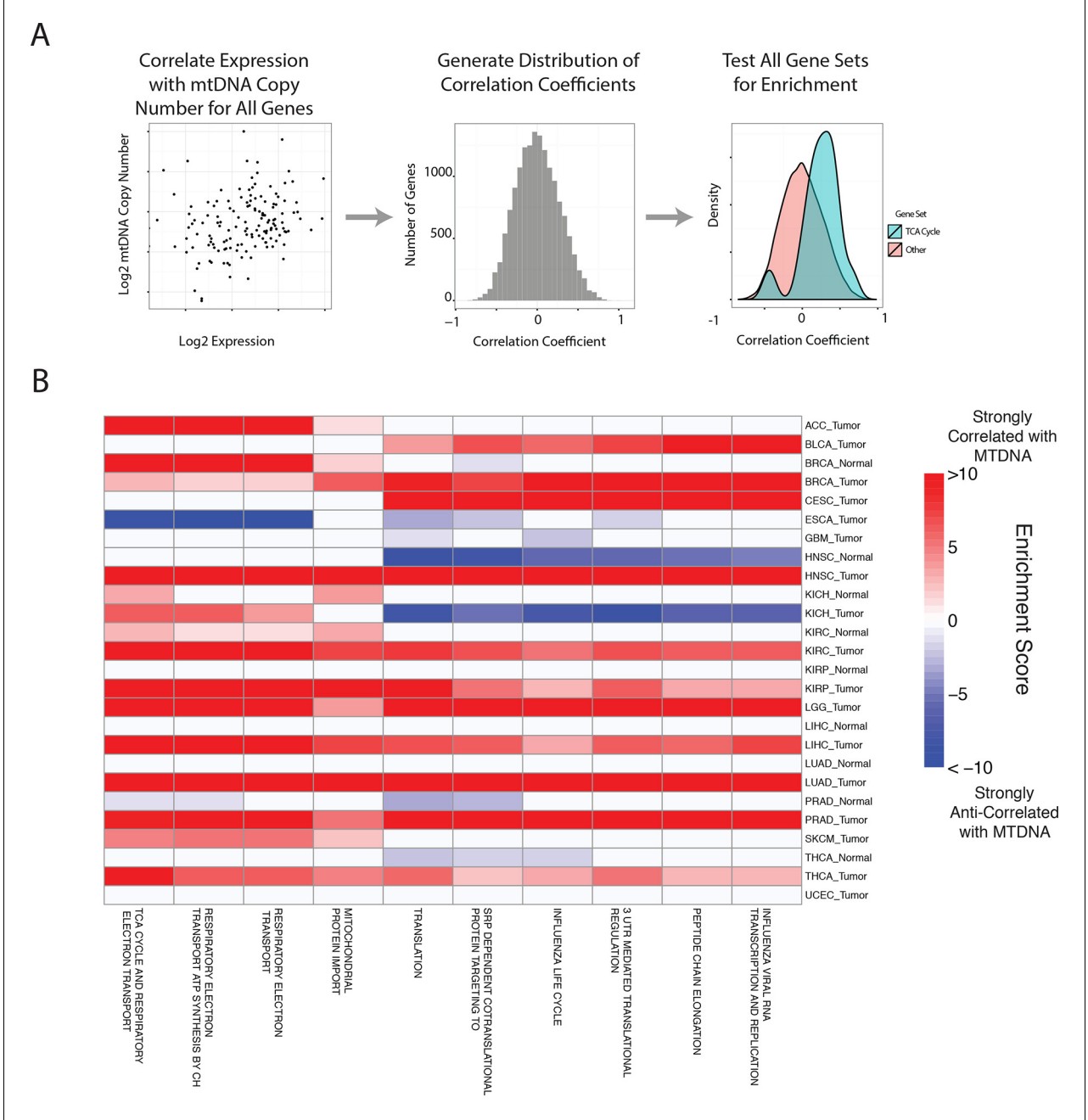

**Figure 5.** Gene set analysis identifies pathways correlated to mtDNA content. (**A**) Correlations between all genes and mtDNA content are calculated. Then, gene sets enriched for high/low correlation coefficients are identified. (**B**) mtDNA copy number is most strongly correlated to metabolic pathways including respiratory electron transport and the TCA cycle, which are localized to the mitochondria. Enrichment score corresponds to the -$\log_{10}$ p-value of the statistical enrichment test, accounting for the sign of the correlation (i.e. positive or negative correlation). Red blocks indicate an enrichment for positive correlation, blue blocks indicate an enrichment for negative correlation. The top ten most frequently positively correlated gene sets across all studies are depicted. Full results are available in ***Supplementary file 2***.

The following source data and figure supplement are available for figure 5:

**Source data 1.** Correlation of mtDNA copy number estimates from WXS and expression of TFAM.

**Figure supplement 1.** The top ten gene sets most frequently negatively correlated with mtDNA copy number across all studies are depicted.

analysis to whole-exome sequencing data and which were not under embargo by the TCGA as of March 2015. All results for the analysis are reported in *Figure 6* and *Supplementary files 3* and *4*.

The most apparent result of our analysis was the association of a large number of CNAs in endometrial carcinomas (UCEC) with increased mtDNA abundance. Recent work by the TCGA proposed a subtype stratification of endometrial carcinomas based on mutation and CNA frequency (*Kandoth et al., 2013*). Among these subtypes is a serous-like 'copy-number-high' subtype with large numbers of somatic CNAs. We obtained the UCEC subtype classifications and confirmed that serous-like endometrial carcinomas exhibited substantially higher mtDNA copy number than all other subtypes (Mann-Whitney p-value $7 \times 10^{-6}$, *Figure 6*), explaining the large number of associations we observed. TP53 mutations are enriched in the serous-like subtype, and these mutations also showed statistically significant association with mtDNA abundance (BH-corrected p-value 0.012).

After removing associations in UCEC, we were left with a small number of statistically significant mutations and CNAs associated with mtDNA abundance. Among these, the strongest signal arose from increased tumor mtDNA content in IDH1-mutant low grade gliomas (*Figure 6*, BH-corrected p-value 0.012). Both IDH1 and IDH2 activating mutations induce production of the so-called 'onco-metabolite' 2-hydroxyglutarate, which competitively inhibits $\alpha$-ketoglutarate-dependent histone demethylases and 5-methylcytosine hydroxylases, inducing a hypermethylation phenotype (*Turcan et al., 2012*; *Xu et al., 2011*). Surprisingly, IDH2 mutations showed no statistically significant change in mtDNA abundance, suggesting that the effect is specific to the cytosolic isoform IDH1. Notably, mutations in PTEN were associated with a significant decrease in mtDNA abundance (BH-corrected p-value 0.033). These results echo a complementary finding by Navis and colleagues (*Navis et al., 2013*), who reported that a mutant IDH1 R132H oligodendroglioma xenograft model displayed high densities of mitochondria and increased levels of mitochondrial metabolic activity. They proposed that an increase in mitochondrial mass would increase activity of mitochondrial IDH2 and compensate for loss of activity introduced by mutant IDH1.

Finally, prompted by a recent report implicating mutations in mtDNA itself with the pathology of kidney chromophobe carcinomas (KICH) (*Davis et al., 2014*), we investigated the connection between mtDNA copy number and mtDNA mutations in KICH. Using somatic mtDNA mutation calls provided by the TCGA (*Davis et al., 2014*), we examined whether mtDNA-mutated samples were likely to have more or fewer mtDNA copies than unmutated samples. We found that samples with mtDNA indels contained much higher quantities of mtDNA than unmutated samples (Mann-Whitney U-test p-value 0.002, *Figure 6—figure supplement 1*). The same effect was not found when examining only single nucleotide variants. These results suggest that the presence of inactivating mtDNA mutations may induce increased mtDNA replication, perhaps as a response to inadequate mitochondrial energy production.

## Immunohistochemical investigation of respiratory protein content

So far, our findings have indicated that a number of tumor types appear to be depleted of mtDNA relative to normal tissue, and that in some (but not all) cases, the amount of mtDNA in a sample is correlated to the expression of respiratory genes. However, in some cancer types (e.g. bladder), tumors exhibited depletion of mtDNA (*Figure 3*), but expression of mitochondrial genes was not correlated to mtDNA copy number (*Figure 5*). This discrepancy is reminiscent of prior work describing mtDNA depletion which was not accompanied by a drop in respiratory activity or mitochondrial protein expression. Instead, a compensation of respiratory activity was described in cases of mtDNA depletion caused by either genetic alterations (*Seidel-Rogol and Shadel, 2002*; *Barthélémy et al., 2001*; *Dorado et al., 2011*) or reverse-transcriptase inhibitors (*Kim et al., 2008*; *Miró et al., 2004*; *Stankov et al., 2007*).

To investigate whether mtDNA depletion was associated with a concurrent decrease of mitochondrial protein expression, we examined the abundance of a mitochondrial protein using immunohistochemistry (IHC) (Thermo Fisher Scientific Mitochondria Ab-2, Clone MTC02, see Materials and methods) in 3 tumor/normal pairs of clear-cell renal cell carcinoma, papillary renal cell carcinoma, and high-grade muscle-invasive urothelial bladder carcinoma (corresponding to TCGA studies KIRC, KIRP, and BLCA respectively; *Figure 7*, and *Figure 7—source data 1*). In KIRC, which was the most strongly mtDNA-depleted tumor type in *Figure 3*, we found significant depletion of mitochondrial protein in all tumor samples compared to adjacent normal renal parenchyma. In KIRP, for which 69% of paired samples were depleted of mtDNA in *Figure 3*, we observed a more subtle

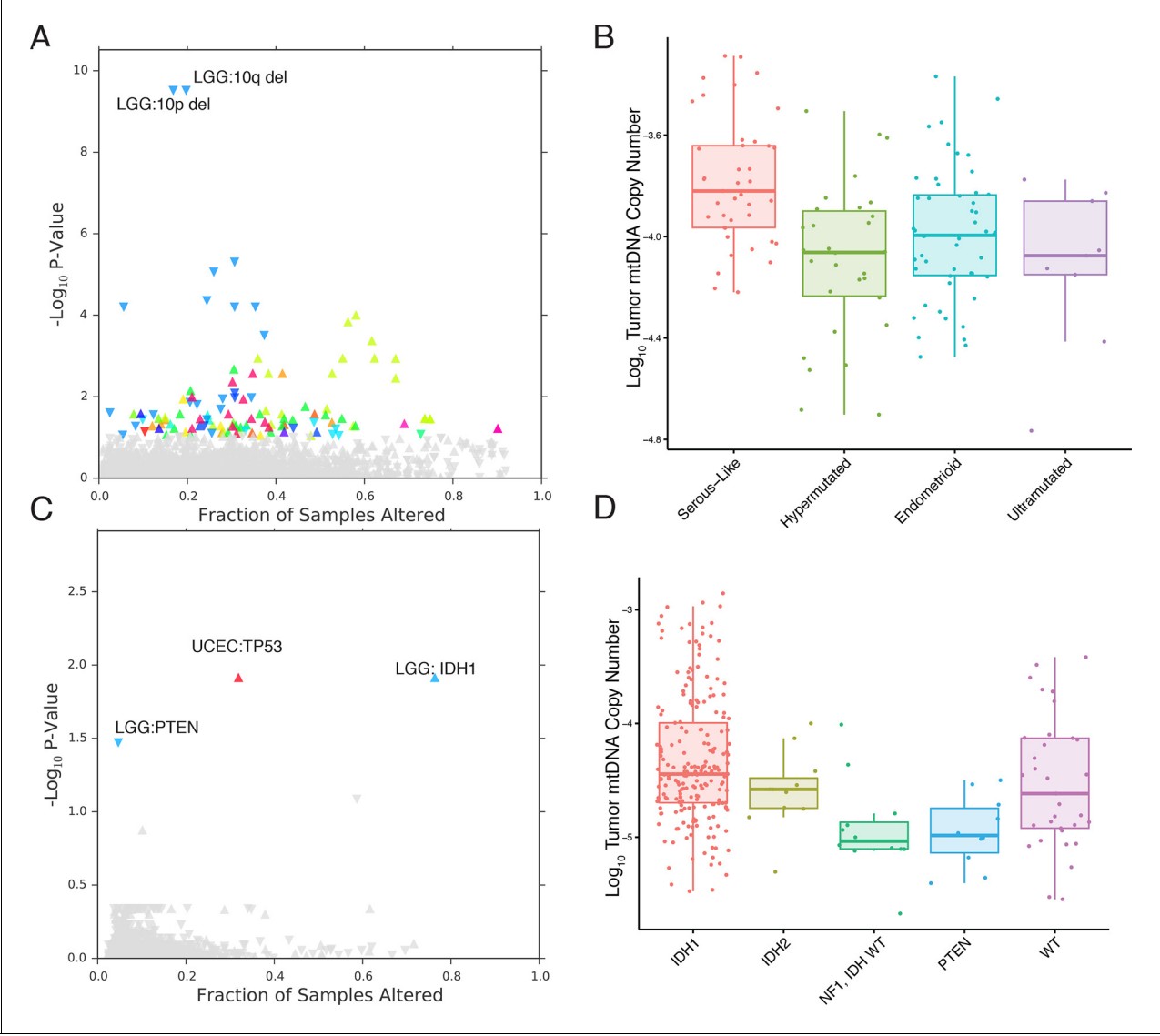

**Figure 6.** mtDNA content is correlated to the incidence of certain mutations and copy number alterations. Each point corresponds to a single alteration (e.g TP53 mutation). Direction of arrow indicates whether alteration increases or decreases mtDNA content. X-axis in (A) and (C) indicates the fraction of samples in a cancer-type that contained the alteration (i.e., ≈ 20% of LGG samples were 10q deleted). (A) 73 out of 1896 copy number alterations (CNAs) tested were found to be significantly associated with mtDNA content (Mann-Whitney p-value <0.05). CNAs from the UCEC cancer type were excluded because of strong association between mtDNA copy number and the 'copy-number high' serous-like subtype in UCEC, shown in (B). (B) The UCEC serous-like subtype displays a marked increase in tumor mtDNA copy number, relative to endometrial tumors of other subtypes. (C) Relatively few mutations (3 out of 3954 tested) associate significantly with tumor mtDNA content (Mann-Whitney p-value <0.05). The UCEC associations are likely the result of the correlation to the serous-like subtype. (D) IDH1 and PTEN mutation status is linked to tumor mtDNA copy number in LGG.

The following figure supplement is available for figure 6:

**Figure supplement 1.** mtDNA copy number levels in kidney chromophobe carcinoma are elevated in samples with truncating mtDNA mutations.

depletion of mitochondrial protein in 2/3 tumor samples, compared to adjacent normal renal parenchyma. In BLCA, we found that 2/3 BLCA tumors showed increased levels of mitochondrial protein, which contrasted with *Figure 3*, where nearly all samples showed evidence of mtDNA depletion.

Collectively, our results from IHC regarding mitochondrial protein expression agree with those from sequencing in 2/3 cancer types (KIRC and KIRP). In a third cancer type (BLCA), mtDNA depletion as quantified by sequencing is not mirrored by a synchronous down-regulation of mitochondrial

protein levels. As mentioned earlier, our results from gene expression analysis (*Figure 5*) indicate that mtDNA copy number is correlated to mitochondrial respiratory gene expression in KIRC and KIRP, but not BLCA. In fact, in BLCA, the gene sets most strongly correlated to mtDNA copy number were associated with the cell cycle and immune response. This suggests that other mechanisms compensate for the depletion of mtDNA in BLCA (and potentially in other cancer types), which is further discussed in the concluding section. Taken together, these results support the notion that factors besides mtDNA copy number can determine the rate of mitochondrial transcription, and that mtDNA depletion is not sufficient evidence to conclude that mitochondrial respiration is down-regulated in a tumor.

## Discussion

In this study, we have investigated the variation of mtDNA copy number levels across many tumor types, arriving at several intriguing observations. Across nearly half of the tumor types we studied, we found evidence for depletion of mtDNA, relative to adjacent normal tissues. Orthogonal measurements of transcription levels (via RNA-Seq) and mitochondrial protein levels (via IHC) in a subset of these samples linked this variation to downregulation of mitochondrially-localized metabolic pathways, in some but not all tumor types.

Our findings of gross changes in mtDNA content in tumors echo a number of prior but isolated observations, largely based on quantitative PCR measurements and with substantially smaller sample sizes, of mtDNA copy number changes in cancers (see [*Yu, 2011*] for a thorough review). For example, oncocytomas (not analyzed in this work) are well-known to be characterized by the excessive accumulation of mitochondria (*Tickoo et al., 2000*). Furthermore, decreases in mtDNA copy number have been reported in breast cancer (*Mambo et al., 2005*; *Fan et al., 2009*), liver cancer (*Lee, 2004*), and clear-cell kidney cancers (*Meierhofer et al., 2004*; *Nilsson et al., 2015*). While the majority of our observations agree with prior work (when comparing to [*Yu, 2011*]), some of our results are in contradiction to prior studies. The discordance between findings seems in part due to inadequate sample sizes, and incomplete or unavailable matched normal tissue. For example, in contrast to (*Mambo et al., 2005*) and (*Wang et al., 2005*), we find no clear increase or decrease in mtDNA content in thyroid or endometrial carcinomas, respectively. However, (*Mambo et al., 2005*) profiled 20 paired thyroid tumors, versus 66 paired thyroid tumors in this report; and (*Wang et al., 2005*) utilized unpaired samples of tumor and normal endometrial tissue (*Wang et al., 2005*), versus 32 paired samples here.

We further showed that mtDNA ploidy alone cannot be used as a surrogate for the respiratory activity of a tumor sample. The literature contains several reports of mtDNA copy number depletion without reduction in mitochondrial transcription/respiratory activity, both in vitro and in vivo. In (*Seidel-Rogol and Shadel, 2002*), HeLa cells depleted of mtDNA by culture in ethidium bromide showed substantial mitochondrial transcription despite the fact that mtDNA, TFAM, and mitochondrial RNA polymerase were all at depleted levels. There, the authors suggest that an excess of TFAM and mitochondrial RNA polymerase prior to depletion may ensure that, even once depleted, transcription is sustained. Another report examined mtDNA depletion as a result of thymidine kinase 2 deficiency in mice, and observed a down-regulation of the mitochondrial transcriptional terminator MTERF3 in heart tissue. As a result, the expression of mitochondrial transcripts (ND6 and COX1) increased in heart tissue, as did the ratio of the levels of these transcripts to mtDNA levels. The consequence of this transcriptional compensation was that the heart tissue was spared from respiratory deficiency (*Dorado et al., 2011*). In tandem with our report, these findings emphasize a nuanced connection between mtDNA copy number and respiratory gene expression. We would argue strongly that future studies investigating changes in mtDNA in tumors should quantify mtDNA protein expression in parallel with estimating mtDNA copy number. A number of related open questions remain to be resolved, including what mechanisms determine the incidence and/or extent of compensation to mtDNA depletion, and what the consequences of mtDNA depletion may be when such compensation takes place (e.g. upregulation of the immune response).

While mtDNA depletion or accumulation may typify certain cancer types, we further identified that subsets of patient samples, characterized by the presence of particular somatic mutations/copy number alterations, were enriched/depleted in mtDNA. The presence of activating IDH1 mutations (in low grade gliomas) or a large number of copy number alterations (in serous-like endometrial

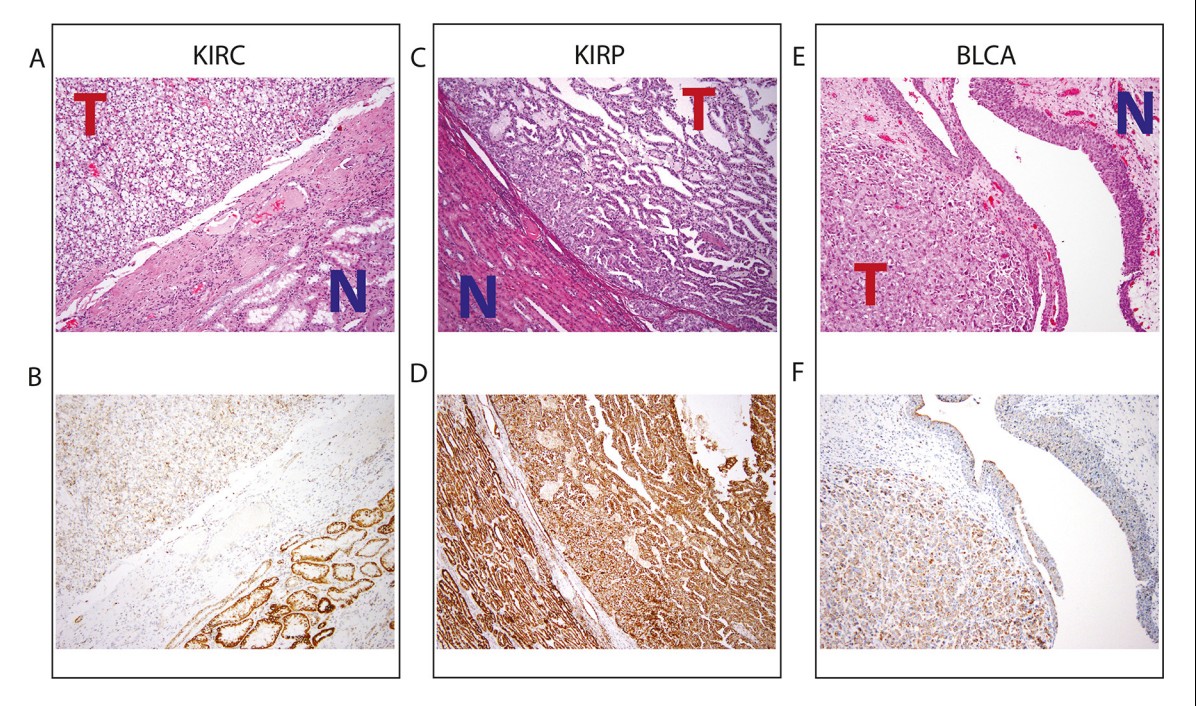

**Figure 7.** Top panel depicts H&E stains, and bottom panel depicts immunohistochemistry with antibody against mitochondrial protein. In all H&E stains, red 'T' indicates tumor tissue, while blue 'N' indicates normal tissue. Orientation of tumor/normal tissue is mirrored in bottom panel. (**A**) H&E-stained section shows clear cell renal cell carcinoma (top left, KIRC Sample 1 from *Figure 7—source data 1*) with the classical features of tumor nests with clear cytoplasm, separated by intricate, branching vascular septae, and adjacent non-neoplastic renal parenchyma (lower right). (**B**) KIRC Sample 1 immunohistochemical staining with MITO Ab2 antibody reveals markedly lower mitochondrial content (cytoplasmic, brown granular positivity) in clear cell RCC compared to normal tubules. (**C**) H&E-stained section shows papillary renal cell carcinoma type 1 (KIRP Sample 3) with tumor (top right) and normal tubules (lower left). (**D**) KIRP Sample 3 immunohistochemical stain with MITO Ab2 antibody shows KIRP with a slightly weaker positivity compared to normal tubules. (**E**) H&E-stained section showing invasive high grade urothelial carcinoma (lower left) with sheets of tumor cells in the lamina propria and the overlying normal urothelium (top right). (**F**) Immunohistochemical staining with MITO Ab2 antibody reveals slightly higher mitochondrial staining in urothelial carcinoma compared to normal urothelium.

The following source data is available for figure 7:

**Source data 1.** Table of results of immunohistochemistry for mitochondrial protein.

carcinomas) is strongly correlated to high tumor mtDNA content. If these tumors (and others with increased mtDNA content) have an increased dependence on mitochondrial metabolism to proliferate, using mitochondrially-targeted therapies (e.g. metformin) may be a therapeutic opportunity. Similarly, vulnerability to mitochondrially-targeted therapies might arise from disabling passenger mutations in genes required for mtDNA copy number maintenance (e.g. DNA polymerase gamma). Both hypotheses should be amenable to investigation in carefully chosen cell line models of cancer.

A number of reports have now described extensive genetic heterogeneity of some tumor types (e.g. kidney cancers [*Gerlinger et al., 2012*]), where spatially distinct biopsies isolated from the same patients have non-overlapping somatic alterations. However, no reports have examined how mitochondrial DNA mutations and copy number vary spatially across a tumor. Variation of this kind, if it exists, might reflect functional diversity in mitochondrial metabolic activity and signaling in different regions of a tumor. Alternately, it would be of particular interest to trace the time-evolution of mtDNA content in a single patient over the course of treatment. As critical players in immunity, signaling, and metabolism, we suspect that mitochondria will inevitably play a role in the evolution of resistance to therapeutic intervention.

## Materials and methods

### Data acquisition

Whole exome sequencing (WXS) and whole genome sequencing (WGS) BAM files for 22 distinct TCGA studies were obtained from the TCGA CGHub repository (*Figure 2*) (*Wilks et al., 2014*). We restricted our analyses to sequence data aligned to GRCH37 using the mitochondrial Cambridge Reference Sequence (CRS). We focused only on primary tumor, adjacent normal tissue, and normal blood samples ('01', '11', and '10' in the sample type field of the TCGA barcode). We further restricted our analyses to samples which were not whole-genome amplified prior to sequencing (i.e., we only used samples containing 'D' in the analyte field of the TCGA barcode), because such amplification could potentially bias the relative abundances of mitochondrial and nuclear DNA in the sample.

Samtools (*Li et al., 2009*) was used to extract reads aligning to the mitochondrial genome meeting the following critieria: (1) passed quality-control, (2) were not marked as duplicate reads, (3) were properly paired, and (4) were aligned with Phred-scaled mapping quality (MAPQ) >30. The number of such reads aligning to the mitochondrial genome was compared to the number of such reads aligning to the nuclear genome.

The pipeline described above includes a number of controls to ensure that mtDNA copy number estimates are not influenced by nuclear integrations of mitochondrial sequences (NUMTs) (*Hazkani-Covo et al., 2010*). A direct result of restricting analysis to properly paired reads is that reads whose mate mapped to a different chromosome are removed prior to copy number calculation. Furthermore, by requiring a conservative Phred-scaled minimal mapping quality of 30 (equivalent to a 99.9% likelihood that reads are aligned to the correct genomic location), reads with homology to nuclear-encoded NUMTs are removed prior to copy number calculations. Prior work has established that more lenient mapping quality thresholds of 20 are sufficient for accurately calling mtDNA copy number (*Ding et al., 2015*)

A complete list of all copy number estimates is available in *Supplementary file 1*.

### Purity and ploidy calculation and correction

Affymetrix SNP6 arrays for tumor and normal samples were acquired for 22 cancer types from the TCGA. Arrays for each individual cancer type were processed together, quantile-normalized and median polished with Affymetrix power tools using the birdseed algorithm to obtain allele-specific intensities. PennCNV (*Wang et al., 2007*) was used to generate log R ratio and B-allele frequencies for each tumor. ASCAT (*Van Loo et al., 2010*) was used to generate allele-specific copy number and estimate tumor ploidy and purity using matched arrays from tumor and normal tissue.

In order to estimate mtDNA copy number in *Equation 1*, we compared the number of reads aligning to the mitochondrial genome to the number of reads aligning to a genome of known ploidy. For samples of normal tissue, we assumed this known ploidy was equal to 2. For tumor tissue which may be infiltrated by stromal/immune cells and copy-number altered, we need to correct for the 'effective ploidy' of the sample. We define this correction factor to be

$$R_{\text{Tumor}} = \frac{\text{Purity} \times \text{Ploidy} + (1 - \text{Purity}) \times 2}{2} \tag{3}$$

where the purity and ploidy values are obtained from ASCAT, as described above. When a sample is composed of pure normal tissue, $R$=1.

### Correction for sequencing center and plate ID

Inspection of mtDNA copy number results indicated a potential association between mtDNA copy number and processing batch. This is consistent with prior reports, e.g. (*Ju et al., 2014*), which described large variation in efficiency of mtDNA depletion in exome sequencing in a sequencing-center-dependent manner. We separately examined the $\log_{10}$ mtDNA copy number for each TCGA plate ID for (1) blood, and (2) tissue-derived (tumor and adjacent-normal tissue) samples. Kruskal-Wallis tests using either blood or tissue-derived mtDNA copy number indicated significant differences in median mtDNA copy number between TCGA plates in 21/22 whole exome sequencing (WXS) datasets (p-value <0.05). In contrast 3/10 whole genome sequencing (WGS) datasets showed significant differences between TCGA plates using tissue-derived mtDNA copy number (5/10 using blood-

derived mtDNA copy number). Manual inspection further indicated that the magnitude of the batch effect was smaller in WGS compared to WXS.

We also calculated, for each TCGA plate $i$ in a given cancer type, the mean mtDNA copy number in (1) blood ($m_i^b$) and (2) tumor/adjacent-normal tissue ($m_i^t$). We observed a statistically significant positive linear correlation (Pearson p-value <0.1 ) between $m^t$ and $m^b$ in 17/19 cancer types profiled with WXS, but in 0/7 cancer types profiled with WGS (analysis restricted to studies with adequate numbers of samples, defined as at least 3 different TCGA plate IDs with at least 3 blood and 3 tissue samples). Importantly, in many cancer types, tumor and blood from the same patient were often processed in different TCGA plates (e.g. 30% in BLCA, 30% in BRCA, and 48% in STAD), and there was no reason a priori to expect that blood and tumor mtDNA estimates should be correlated across batches. The existence of such a correlation suggested a TCGA-plate-dependent contribution to the observed copy number of all samples (both blood and tissue-derived) processed within that plate.

Taken together, the results above suggested that a batch effect associated with TCGA plate ID/sequencing center was present in WXS, and possibly also in WGS. Based on these considerations, the small magnitude of the batch effect in WGS, and the prior literature describing significant variation in mtDNA sequencing depth in exome sequencing due to differences in exonic enrichment (*Ju et al., 2014*), we elected to control for a batch effect in WXS, but not WGS. Importantly, in *Supplementary file 1*, we report uncorrected mtDNA copy number, corrected mtDNA copy number, and plate IDs for all samples analyzed, so that future work by others may model the effect in whatever manner they see fit.

The set of observed $\log_{10}$ mtDNA copy numbers $M$ are indexed by the TCGA plate $i$, the tissue type (either tumor/adjacent-normal or blood) $j$, and the sample $k$. We fit the following linear model to the mtDNA copy numbers $M_{ijk}$:

$$M_{ijk} = \mu + \sum_{plate} \alpha_i P_i + \sum_{tissue} \beta_j T_j + \epsilon_{ijk} \tag{4}$$

where $P_i$ and $T_j$ are indicator variables corresponding to the plate and tissue that a sample comes from. The parameter $\mu$ is the grand mean of mtDNA copy number, $\alpha$ the effect attributable to the plate and $\beta$ that attributable to the tissue. The residual $\epsilon$ is the sum of the true $\log_{10}$ mtDNA copy number and measurement error. We constructed a linear model corresponding to *Equation 4* for each cancer type. Then, we corrected the observed $\log_{10}$ mtDNA copy number for each WXS sample according to the contribution $\hat{\alpha}$ from the corresponding TCGA plate ID. Corrected copy number values were used for subsequent survival, gene expression, and somatic alteration analysis.

## Survival analysis

Survival analysis was performed with univariate Cox proportional hazards regression models where the independent and dependent variables were the $\log_{10}$-transformed corrected mtDNA copy number and the overall survival respectively. The p-values for the significance of mtDNA copy number as a predictor of survival were obtained from Wald tests.

## Gene set analysis

RSEM normalized RNA-Seq gene expression data were downloaded from the Broad Firehose, using the most recent data as of November 4, 2014. Data were filtered to remove genes with average read count less than 16. We calculated the nonparametric Spearman correlation and associated p-value between the expression of each gene and the copy number of mtDNA. To remove putatively spurious correlations, we identified all genes with a p-value greater than 0.05, and set the correlation coefficient for those genes equal to zero. We then use the geneSetTest function in the limma (*Law et al., 2014*) package to test whether particular gene sets showed an enrichment for positive/negative correlations, relative to the distribution of correlations across all genes. Thus, a single p-value was calculated for each alternative hypothesis. All p-values were adjusted using the Benjamini-Hochberg procedure. The method described above was applied to every Reactome pathway in the MSigDB canonical pathways gene set. The enrichment score depicted in *Figure 5* is the $-\log_{10}$ corrected p-value of this statistical test, with the color (red or blue) indicating the direction of enrichment.

The analysis was run separately for tumor and normal tissues. We applied our gene set analysis pipeline to all studies for which we had at least 20 samples of RNA-Seq data (in order to retain sufficient statistical power). Analyses were run for each combination of tumor type and tissue, and ensuing results were then aggregated across all studies. All results from the analyses are provided in the *Supplementary file 2*.

## Mutation and copy number alteration analysis

For each study, Gistic2 and MutSigCV results were downloaded from the Broad Firehose (most recent data as of Nov 14, 2014). From Gistic, we retained all arm-level and focal alterations with q-value less than 0.1. For mutations, we obtained the MAF file from the output of MutSig. For each gene, we calculated the number of patients in which this gene exhibited a nonsynonymous, coding mutation (i.e., missense, non-sense, frameshift, in-frame insertion/deletions, and splice-site mutations), excluding those with greater than 600 non-synonymous coding mutations). We then retained any genes which were mutated in greater than 4% of patients. Non-parametric Mann-Whitney U-tests were used to evaluate whether tumors bearing a particular somatic alteration contained significantly higher/lower amounts of mtDNA in tumor samples. After testing all associations, p-values obtained from the U-tests were corrected using the Benjamini-Hochberg procedure.

## Histology

All tissues were fixed in 10% neutral-buffered formalin and paraffin embedded as part of a routine surgical pathology procedure and 5-micron-thick sections stained with Hematoxylin and eosin (H&E) were reviewed. Immunohistochemical (IHC) analysis was performed on 5-micron-thick sections by Ventana, Discovery XT immunohistochemical stainer. The sections were deparaffinized and subjected to heat induced antigen retrieval using CC1 at high pH before primary incubation with MITO Ab2 (mouse monoclonal, clone MTC02, Neomarkers, 1:50 dilution). Slides were then counterstained with hematoxylin, dehydrated and cover-slipped.

## Acknowledgements

We thank Deborah S Marks, Nick Gauthier, Arman Aksoy, Nils Weinhold, and Alessandro Pastore for thoughtful discussions and feedback.

## Additional information

### Funding

| Funder | Grant reference number | Author |
| --- | --- | --- |
| National Institutes of Health | 5U24 CA143840-05 (Sander) | Eduard Reznik<br>Yasin Şenbabaoğlu<br>Chris Sander |
| National Institutes of Health | P30 CA008748 | Ed Reznik<br>Martin L Miller<br>Yasin Şenbabaoğlu<br>Nadeem Riaz<br>Judy Sarungbam<br>Satish K Tickoo<br>William Lee<br>Venkatraman E Seshan<br>A Ari Hakimi<br>Chris Sander<br>Hikmat A Al-Ahmadie |

The funders had no role in study design, data collection and interpretation, or the decision to submit the work for publication.

### Author contributions

ER, Conception and design, Acquisition of data, Analysis and interpretation of data, Drafting or revising the article; MLM, YŞ, NR, JS, SKT, WL, VES, AAH, CS, Analysis and interpretation of data,

Drafting or revising the article; HAA-A, Analysis and interpretation of data, Drafting or revising the article

**Author ORCIDs**
Ed Reznik, http://orcid.org/0000-0002-6511-5947
Yasin Şenbabaoğlu, http://orcid.org/0000-0003-0958-958X

# Additional files

**Supplementary files**

• Supplementary file 1. Summary table of mtDNA copy number in tumor, adjacent-normal, and blood samples from the TCGA. Data for a patient is included if and only if a tumor sample was sequenced. Normal tissue/blood data without a matching tumor sample is not included, but was used for batch-correction calculations.

• Supplementary file 2. Results of gene set analysis. Enrichment scores for each cancer type are negative log10 p-values.First column indicates enrichment score for positive correlations between mtDNA copy number and gene expression, second column indicates enrichment score for negative correlations between mtDNA copy number and gene expression.

• Supplementary file 3. Results of association analysis with copy number alterations. As mentioned in the main text, associations with the UCEC cancer type are removed.

• Supplementary file 4. Results of association analysis with mutations.

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
