## [Decision Letter]

Thank you for submitting your work entitled "Mitochondrial DNA Copy Number Variation Across Human Cancers" for peer review at *eLife*. Your submission has been favorably evaluated by Randy Schekman as Senior editor and two reviewers, one of whom, Chi Dang, is a member of our Board of Reviewing Editors.

The reviewers have discussed the reviews with one another and the Reviewing editor has drafted this decision to help you prepare a revised submission.

Summary:

The manuscript by Reznik et al. reports a comprehensive informatic analysis of mitochondrial DNA content in ~13,000 tumor samples from the TCGA effort. This study is potentially of high significance as it addresses systematically mtDNA copy number changes in various solid tumors for the first time. Other studies have done this before, but usually only on one tumor type and often with not enough samples to draw strong conclusions. This remarkable study uncovers the broad depletion of mtDNA in tumor vs normal samples across many, but not all tumor types. This study goes further to try and relate mtDNA copy number changes to gene expression and mutational changes, which is lauded and interesting. Informatic analyses also reveal that mtDNA content correlates with specific changes such as IDH1 mutation. Further, mtDNA content correlates with clinical outcome for certain tumor types. If the mtDNA copy number changes found are indeed valid, the study does break interesting new ground and moves the field forward.

Essential revisions:

1) There are hundreds of insertions of mtDNA sequences in the nuclear genome (so called NUMTs), some are even full-length mtDNA insertions. The authors do not mention these or acknowledge that they would be incorrectly assigned as actual mtDNA reads in the whole-genome sequencing method of mtDNA copy number. This may lead to false positive results regarding mtDNA copy number in the tumors and need to be subtracted from the analysis.

2) The ASCAT algorithm seems quite good at factoring out ploidy differences, but less robust with regard to the purity issue. Thus, there remains concern that the differences in mtDNA copy number observed somehow reflects the different cell composition of the tumor versus control normal tissue. For example, immune and stromal cells might have lower basal mtDNA copy number than resident tissue. The tumor samples assayed almost certainly have a high percentage of these cell types compared to normal tissue and hence the trend toward depletion in many of the tumor samples may simply reflect this. Can the authors provide some histology to bolster their conclusions? It is too much to do this for all the samples, but one or two representative cases to support their conclusions seem necessary.

3) TFAM is an abundant mtDNA-binding protein and transcription/replication factor that often correlates with mtDNA copy number. Can the authors address whether mutations, gene expression changes, or amounts of TFAM correlate with the mtDNA copy number changes they report?

4) Recently mtDNA stress (depletion and TFAM reductions) has been shown to activate innate immune pathways (e.g. interferon stimulated genes) (West et al. Nature 2015). Is there a gene expression profile indicative of this in the tumors that show mtDNA depletion?

[Editors' note: further revisions were requested prior to acceptance, as described below.]

Thank you for resubmitting your work entitled "Mitochondrial DNA Copy Number Variation Across Human Cancers" for further consideration at *eLife*. Your revised article has been favorably evaluated by Randy Schekman as Senior editor and two reviewers, one of whom is a member of our Board of Reviewing Editors. The manuscript has been improved but, as requested, we are granting you an extra month for the additional data analysis.

---

## [Author Response]

Summary:

*The manuscript by Reznik et al. reports a comprehensive informatic analysis of mitochondrial DNA content in ~13,000 tumor samples from the TCGA effort. This study is potentially of high significance as it addresses systematically mtDNA copy number changes in various solid tumors for the first time. Other studies have done this before, but usually only on one tumor type and often with not enough samples to draw strong conclusions. This remarkable study uncovers the broad depletion of mtDNA in tumor vs normal samples across many, but not all tumor types. This study goes further to try and relate mtDNA copy number changes to gene expression and mutational changes, which is lauded and interesting. Informatic analyses also reveal that mtDNA content correlates with specific changes such as IDH1 mutation. Further, mtDNA content correlates with clinical outcome for certain tumor types. If the mtDNA copy number changes found are indeed valid, the study does break interesting new ground and moves the field forward.*

Thank you. A small comment: we analyze mtDNA copy number in ~13000 tissue samples, of which ~6000 are tumors. The remaining samples are blood or adjacent normal tissue.

Essential revisions:

*1) There are hundreds of insertions of mtDNA sequences in the nuclear genome (so called NUMTs), some are even full-length mtDNA insertions. The authors do not mention these or acknowledge that they would be incorrectly assigned as actual mtDNA reads in the whole-genome sequencing method of mtDNA copy number. This may lead to a false positive results regarding mtDNA copy number in the tumors and need to be subtracted from the analysis.*

We agree with the reviewer. Our computational pipeline includes a number of features that subtract the contribution of nuclear insertions of mtDNA to calculations of mtDNA copy number. These filters are based on those implemented by others in prior work (e.g.Ding 2015), although we have actually increased their stringency in this work (detailed below). We failed to describe these “hidden features” of our approach in the initial manuscript. We now include a discussion of these filters in the Methods section:

*“*The pipeline described above includes a number of controls to ensure that mtDNA copy number estimates are not influenced by nuclear integrations of mitochondrial sequences (NUMTs) (Hazkani-Covo 2010). A direct result of restricting analysis to properly paired reads is that reads whose mate mapped to a different chromosome are removed prior to copy number calculation. Furthermore, by requiring a conservative Phred-scaled minimal mapping quality of 30 (equivalent to a 99.9% likelihood that reads are aligned to the correct genomic location), reads with homology to nuclear-encoded NUMTs are removed prior to copy number calculations. Prior work has established that more lenient mapping quality thresholds of 20 are sufficient for accurately calling mtDNA copy number (Ding 2015).”

The filter most relevant to the reviewers’ comment is that which requires reads to be properly paired, which ensures that the mate of a read is located on the same chromosome as the read, and within a reasonable distance. We used the samtools flagstat command to confirm that the mates of MT reads were also aligning to the MT genome.

Interestingly, in 7% of samples, we observed an extremely small proportion of MT reads (never more than 0.04%, or 4 out of every 10000, MT reads, and thus *not affecting mtDNA copy number calculations*) that, despite our filters, whose mates mapped to a different chromosome as reported by BWA. Upon further investigation, all such reads appeared to map to the terminal end of the Y chromosome. It appears that the flagging of such reads as properly paired despite mapping to different chromosomes may result from concatenation of the Y and MT chromosome during the indexing of the reference sequence by BWA. *These artifacts affected our results to a completely negligible extent. However, we thought it useful to include this finding in the response document, as a guide for others who may encounter the artifact in future studies*.

*2) The ASCAT algorithm seems quite good at factoring out ploidy differences, but less robust with regard to the purity issue. Thus, there remains concern that the differences in mtDNA copy number observed somehow reflects the different cell composition of the tumor versus control normal tissue. For example, immune and stromal cells might have lower basal mtDNA copy number than resident tissue. The tumor samples assayed almost certainly have a high percentage of these cell types compared to normal tissue and hence the trend toward depletion in many of the tumor samples may simply reflect this. Can the authors provide some histology to bolster their conclusions? It is too much to do this for all the samples, but one or two representative cases to support their conclusions seem necessary.*

The reviewers raise an intriguing point that occurred to us in the course of our investigation. We agree that when calculating tumor mtDNA copy number, we are in fact reporting the average aggregate copy number of the tumor sample, including contributions from stromal and immune cells. With this in mind, we felt that the reviewers’ remark rested on two separate questions:

First, can we provide orthogonal evidence that some tumor types are depleted of mtDNA? Second, more broadly, is mtDNA content associated with infiltration of the tumor by stroma and immune cells?

To address the first question, we completed immunohistochemistry on three clear-cell renal cell carcinoma (KIRC) tumor/normal pairs and three papillary renal cell carcinoma (KIRP) tumor/normal pairs using an antibody for an mtDNA-encoded protein (MTCO2, a component of Complex IV). In both cases, we ensured that adjacent-normal tissue was visible in the field of view.

We remind the reviewers that KIRC was found to be the most strongly mtDNA-depleted tumor type in our analysis, while KIRP was observed to be depleted but to a much lesser extent. As shown in Figure 3—figure supplement 2 and [Supplementary-material SD1-data], we found that all 3 clear-cell tumor showed very poor staining for MTCO2, compared to adjacent normal tissue. In contrast, 2/3 papillary tumors also showed less staining for MTCO2 than adjacent normal tissue, but the magnitude of this effect was much smaller/more subtle than for the clear-cell samples. These findings are in line with those reported in Figure 3.

To address the second question, we obtained estimates of stromal and immune cell infiltration for 8 cancer types calculated using the ESTIMATE algorithm published in 2013. We calculated the correlation between each of [stromal score, immune score], and [tumor mtDNA content, log_2_ ratio of tumor:normal mtDNA content]. These results are reported in Figure 3—figure supplement 3–Figure 3—figure supplement 5, [Supplementary-material SD1-data], [Supplementary-material SD2-data], and [Supplementary-material SD3-data]. In most cases, we found no correlation between mtDNA content and these “purity parameters.” However, there were notable cases where we did observe a weak correlation: for example, bladder, breast, and endometrial tumors exhibit a negative correlation between immune cell infilitration and tumor mtDNA content. Importantly, even in these cases, the correlation between mtDNA content and stromal/immune infilitration is insufficient to explain the broad pattern of mtDNA depletion that we observe. They do, however, raise the interesting possibility that changes in mtDNA content may synchronously drive (or be driven by) changes in immune/stromal content (as evidenced by the relationship between interferon signaling and mtDNA depletion, described in Comment 4 below). We report these findings in the subsection “Gross Changes in mtDNA Content are Evident in Many Cancers”.

3) TFAM is an abundant mtDNA-binding protein and transcription/replication factor that often correlates with mtDNA copy number. Can the authors address whether mutations, gene expression changes, or amounts of TFAM correlate with the mtDNA copy number changes they report?

The reviewers raise an intriguing question, as TFAM is critical both to mtDNA transcription and packaging into nucleoids. As of October 8, 2015, cbioportal.org reported only a handful of TFAM missense/nonsense mutations across all studies in the TCGA. Rather than try to tease apart the contribution of these rare mutations to mtDNA copy number, we instead focused on the association of changes in TFAM gene expression and mtDNA copy number.

After separating cancer studies by sequencing center (e.g. Broad Institute, Baylor) and tumor/normal tissue identity, we computed the correlation between TFAM gene expression and mtDNA copy number for all cases where we had at least 30 samples. The results are reported in Table 5, with additional information on the number of samples available and the Log10 range of mtDNA copy number (i.e. a value of 1 indicates the difference between the maximal and minimal mtDNA copy number was a factor of 10). Of the 29 cases tested, 13 of them (~45%) showed statistically significant positive correlation between mtDNA copy number and TFAM expression (Spearman rho p-value <0.05).

Interestingly, we found that the studies that did not exhibit a positive correlation between mtDNA content and TFAM expression tended to have very little variation in mtDNA content (i.e. the Log10 range was small). In fact, there was a strong correlation between the significance of the correlation (-log_10_(P-value)) and the range of mtDNA content (see Figure 8, Spearman rho p-value 0.0007). Based on this observation, we speculate that the poor correlation between mtDNA content and TFAM expression may, in some cases, arise because of biological effects (e.g.small variation in mtDNA content may not translate to comparable changes in TFAM expression) or technical issues (e.g.limits to the detection sensitivity of our mtDNA copy number calculations).

Author response image 1.**DOI:**
http://dx.doi.org/10.7554/eLife.10769.025

*4) Recently mtDNA stress (depletion and TFAM reductions) has been shown to activate innate immune pathways (e.g. interferon stimulated genes) (West* et al. *Nature 2015). Is there a gene expression profile indicative of this in the tumors that show mtDNA depletion?*

In our initial submission, we briefly mentioned that several of the pathways most negatively correlated with mtDNA content were immune pathways. We revisited and expand on this finding, and now report the results in Figure 5—figure supplement 1. Of the top 10 most frequently negatively correlated pathways, two were associated with interferon signaling. Among the 7 tumor types observed to be depleted of mtDNA relative to normal tissue and with available RNA-Seq data (no RNA-Seq for stomach adenocarcinomas), 6/7 exhibited statistically significant anti-correlation between interferon signaling and tumor mtDNA copy number. In other words, in these tumor types, the interferon signaling pathway was most highly expressed in samples with the lowest mtDNA copy number.

Upon investigating further, we found that the reviewers’ comment had exposed a peculiar and quite interesting pattern in the results of our gene set analysis. For 4 tumor types with depletion of mtDNA and sufficiently numerous adjacent-normal samples (BRCA, HNSC, KIRC, LIHC), the negative correlation between interferon signaling expression and mtDNA content was not present in expression data for normal tissue. In other words, in these tissues, mtDNA content was anti-correlated with interferon signaling in tumor samples, but not in normal samples. This finding is now reported in the subsection “mtDNA Copy Number is Correlated to the Expression of Mitochondrial Metabolic Genes”.

[Editors' note: further revisions were requested prior to acceptance, as described below.]

The manuscript has been improved but, as requested, we are granting you an extra month for the additional data analysis.

Thank you for the opportunity to revise our manuscript. Following submission of our revised manuscript in November 2015, we became aware of a potential batch effect in our mtDNA copy number calculations using whole exome sequencing data. We found that whole exome sequencing batches (defined by the sequencing center and TCGA plate ID) displayed significant variation in the efficiency of targeted exonic enrichment (and, therefore, removal of mtDNA reads). While we had partially controlled for this effect in our initial submission by separating analysis by sequencing center, inspection of the data indicated that controlling for plate ID was also necessary. Importantly, this batch effect was not evident for whole-genome sequencing data, supporting the hypothesis that it arose because of targeted enrichment of exonic regions.

We have completed several revised analyses after controlling for the batch effect, which are detailed in the attached manuscript. In summary, our results remain largely unchanged, and in fact improve in several respects.Changes to prior results are detailed below.

We have also included one additional analysisin the manuscript, which is detailed in Comment 7below. There, we complete additional immunohistochemical staining of a third cancer type (bladder) to examine the effects of mtDNA depletion on respiratory protein expression. Unlike the other mtDNA-depleted cancer types which we have stained (kidney clear-cell and kidney papillary), we found that bladder tumors did not show depletion of MT-CO2 (a mtDNA-encoded protein), suggesting that bladder tumors may compensate for mtDNA depletion to sustain mitochondrial transcription.

1) We implemented a batch correction using a linear model. By jointly modeling the contribution of batch to mtDNA copy number in blood and tissue (i.e.tumor/adjacent-normal), we were able to normalize mtDNA copy number estimates across all batches. A detailed description of the batch effect and our methodology to correct for it are in the subsection “Correction for Sequencing Center and Plate ID”.

2) To facilitate use of our results by others, and to promote transparency, we have augmented our primary data table ([Supplementary-material SD4-data])to include the original (un-corrected) mtDNA copy number estimates, batch IDs, and the newly batch-corrected mtDNA copy number estimates.

3) We repeated the association analysis comparing mtDNA copy number estimates from whole-genome and whole-exome sequencing, controlling for batch. Our results are now significantly improved, and no longer need to be separated by sequencing center. To simplify the presentation, we have removed the table reporting the p-values of the correlations associated with this analysis, and now report them directly in the caption of the figure. All correlations between WXS and WGS are highly significant, and generally improve compared to the prior submission.A comparison of the two results (pre-batch correction vs. post-batch correction) is below. The effect of the correction should be evident when comparing the same study across the two figures (e.g.BLCA, top left corner of both). Please see Figure 2—figure supplement 1 too.

Author response image 2.Comparison of mtDNA copy number estimates from samples profiled using both WGS and WXS.Results above are those prior to batch-correction of mtDNA copy number.**DOI:**
http://dx.doi.org/10.7554/eLife.10769.026

4) We repeated the analysis comparing mtDNA copy number in tumor and adjacent-normal tissues, restricting ourselves to pairs of samples processed in the same batch. We found that ~90% of all tumor-normal pairs were processed in the same batch, and our results are largely unchanged, except that the depletion effect in one cancer type (stomach adenocarcinomas) now loses statistical significance upon multiple hypothesis correction.

5) We repeated analysis associating mtDNA copy number with gene expression data. Our results are largely unchanged, with identical “major findings.”Using batch-corrected mtDNA copy number estimates, more cancer types now exhibit correlation between respiratory gene expression and mtDNA copy number. In terms of expression data, mitochondrial respiratory genes remain the most recurrently correlated gene set with mtDNA copy number. Interestingly, some tumor types which did not show correlation with respiratory genes, now do (e.g.HNSC tumors). Of note, while the anti-correlation between expression of 2 different interferon-related genesets (α/β interferon and total interferon) and mtDNA copy number is still retained, it is no longer in the top-10 most anti-correlated gene sets and does not appear in Figure 5—figure supplement 1. However, the 2 gene sets are still among the most anti-correlated, ranking 27_th_ and 40_th_ respectively, out of 674 total gene sets ([Supplementary-material SD5-data]). Notably, other immune-related pathways remain in the top 10 in Figure 5—figure supplement 1.

We also repeated the analysis comparing mtDNA copy number estimates with ESTIMATE scores in the subsection “Gross Changes in mtDNA Content are Evident in Many Cancers”, Figure 3—figure supplement 2–Figure 3—figure supplement 5, and [Supplementary-material SD2-data]. We observed a weak negative correlation between immune scores in ESTIMATE and mtDNA copy number across several cancer types, which echoed our results from the gene expression analysis described above.

6) We repeated the analysis associating mtDNA content with somatic alterations. The results are qualitatively unchanged, although in some cases p-values did become less significant(i.e. the p-value for the association between IDH1 mutations in gliomas and mtDNA content is now larger/less significant than in prior versions). NF1 mutations are no longer statistically significantly associated with lower mtDNA content because of multiple hypothesis correction (un-corrected p-value 0.006), but we still feature the alteration in Figure 6.

7) A major new analysis in our manuscript is additional immunohistochemical staining of three high-grade muscle-invasive urothelial (bladder) cancers. This addition was not prompted by reviewers’ comments, but rather by our own curiosity and desire to understand a strange, counterintuitive phenomenon in our results. We noted that (1) bladder cancers are depleted of mtDNA content relative to normal tissue both in WXS and WGS (Figure 3), but (2) using gene expression data, levels of mtDNA in bladder tumors are not correlated to expression of respiratory genes (Figure 5). This left us asking whether the depletion in mtDNA in bladder tumors actually translated to a depletion in respiratory protein levels.

As we explain in detail in the subsection “Immunohistochemical Investigation of Respiratory Protein Content” and newly added Figure 7,and discuss in the Discussion, we used immunohistochemistry to stain 3 bladder tumors and adjacent-normal urothelium for MT-CO2. Two of the three cases showed stronger staining in tumor than in normal, although the magnitude of the effect was relatively small compared to the difference in staining in KIRC. These results suggest that there may be a compensatory effect in bladder cancers, such that respiratory capacity is sustained despite reduced mtDNA copy number. As we describe in the newly added text in third paragraph of the Discussion, a number of other groups have reported similar compensation in other instances mtDNA depletion (in vitro, and in vivo).

We feel this new data should be reported in the manuscript. It emphasizes, as one of the reviewers pointed out in an earlier reviewers’ comment, that our observation of mtDNA depletion across many solid tumor types should not be misconstrued for a widespread “Warburg-effect.” It highlights the nuanced regulation of mitochondrial transcription, and it opens several avenues for further research as to what the causes and consequences of mtDNA depletion are.

[Editors' note: During proofing of their manuscript, the authors noticed that they had incorrectly described an antibody using for staining in Figure 7 and asked for this to be corrected. The authors stained with mitochondrial antibody MTC02, which recognizes a 60 kDa mitochondrial protein. In the manuscript and in the response to reviewers, the authors describe this antibody as MT-CO2, and write that it recognizes a mitochondrial DNA-encoded protein. The manuscript has been corrected accordingly.]